# Updated Decision Aid Enabling Women to Choose between with or without Epidural Analgesia during Childbirth, and Confirmation of Validity

**DOI:** 10.3390/ijerph20116042

**Published:** 2023-06-02

**Authors:** Eri Shishido, Yumiko Arabiki, Shigeko Horiuchi

**Affiliations:** 1Graduate School of Nursing Science, St. Luke’s International University, Tokyo 104-0044, Japan; 2Makita General Hospital, Tokyo 144-8501, Japan

**Keywords:** epidural anesthesia, natural birth, decision aid, pregnancy, birth, face validity

## Abstract

Background: The use of a decision aid for choosing whether to have or not have anesthesia during childbirth has been shown to increase both knowledge about birth and the proportion of women who made their own decisions compared with women who did not use a decision aid. Herein, we updated the first version of our decision aid into a second version and evaluated this updated decision aid. We evaluated the face validity and content appropriateness of the updated decision aid developed to enhance the ability of women to choose between childbirth with or without epidural analgesia. Methods: This was a descriptive study based on a literature review of updated information for addition to the first version. PubMed and Cochrane Library were searched from 2003 to May 2021. Thereafter, obstetricians, anesthesiologists, and midwives were asked to respond to a questionnaire regarding the face validity and content appropriateness of the updated decision aid regarding whether it meets the IPDASi (Version 4.0) quality standards. Results: One obstetrician, one anesthesiologist, and three midwives who had performed epidural anesthesia for at least three years responded to the questionnaire. The responses to the evaluation items of face validity (i.e., style and clarity) were positive. There were 38 specific comments regarding content appropriateness classified into seven categories: “addition or revision of text”, “unification of expressions”, “need for explanation/information”, “lack of evidence”, “potential to mislead”, “questionable”, and “structure”. Conclusion: The face validity and content appropriateness of the updated decision aid was confirmed. The next step is evaluation of the updated decision aid by pregnant women who give birth.

## 1. Background

In Japan, there has been an increase in the percentage of pregnant women who choose epidural analgesia for delivery from 4.6% in 2014 to 8.6% in 2020 [1,2]. On the other hand, only 5.9% of health facilities provide information on epidural analgesia for delivery to all pregnant women, whether they wish to have anesthesia or not [3]. Therefore, pregnant women who are undecided about their choice of epidural anesthesia gather information from the Internet, friends and acquaintances, consult with doctors and midwives at antenatal check-ups, and attend classes on epidural anesthesia. Based on the information they have gathered, they consider the advantages and disadvantages of having epidural anesthesia and then decide whether to have one or not.

Shishido et al. (2018) [4] previously reported the birth outcomes of undecided pregnant women in Japan who were divided between epidural anesthesia and natural birth. Pregnant women who were undecided regarding their birth method had a lower mean age and were composed of a larger proportion of primiparous mothers than women who had decided between epidural anesthesia and natural birth. This suggested that Japanese pregnant women had difficulty deciding on the birth method because of their lack of knowledge and information about birth [4]. Thus, a decision-making aid for choosing between having epidural anesthesia or natural birth was developed in Japan by Henna et al. (2019) [5]. The developed decision aid with Japanese and English versions was published in Ottawa’s A to Z Inventory of Decision Aids (https://decisionaid.ohri.ca/AZsumm.php?ID=1992, accessed on 12 December 2021) and made readily available to many people.

In their systematic review, Stacey et al. (2017) [6] reported that the use of decision aids when making choices could improve knowledge about different options, make it easier to recognize risks, reduce conflicts caused by lack of information or ambiguity about values, and clarify what is most important to oneself and what values are important.

Shishido et al. (2020) [7] previously evaluated approximately 300 pregnant women and found that the intervention group (i.e., used the developed decision aids) had significantly fewer pregnant women who were undecided about their choice of birth method than the control group (i.e., used a general pamphlet). Additionally, the changes in pre-test and post-test knowledge scores on a 10-point knowledge test including the benefits and risks of epidural anesthesia were significantly higher in the intervention group using the decision aids than in the control group (difference in knowledge scores; intervention group: 1.96 [SD 1.63], control group: 1.33 [SD 1.44], *p* < 0.001). The results are similar to those of Stacey et al. (2017) [6], suggesting that decision aids could be useful in supporting pregnant women to make a choice between having epidural anesthesia or natural birth. Ito (2016) [8] identified the information pregnant women wanted to know about epidural anesthesia, namely, the effects of anesthesia on the child (18.3%), the effects of anesthesia on birth (16.0%), the cost of anesthesia (16.0%), the experience of other women (7.1%), the facility performance of epidural analgesia (4.7%), the possibility of cesarean section (4.1%), and the facility performing the procedure (4.1%).

However, the first version of our decision aid focused heavily on the birth outcomes of epidural anesthesia and natural birth. Therefore, we considered herein the need for a new decision aid containing information on the needs of pregnant women and social concerns, such as the condition of the child after birth, the status of breastfeeding, and the impact of birth on postnatal mental health.

Based on the above perceived need, a second version of the aid prototype was developed by updating and adding information to the first version of the decision aid developed by Henna et al. (2019) [5] necessary to support decision making. The purpose of the present study was to examine the face validity and the appropriateness of the content of the updated decision aid.

## 2. Methods

### 2.1. Study Design

We used a descriptive study design to evaluate the face validity and content appropriateness of the updated decision aid.

### 2.2. Process for Developing the Updated Decision Aid

#### 2.2.1. Search by Keyword

Searches were conducted in Cochrane Library and PubMed for international reports that will be used for updating information in the first decision aid and for adding new information in the second decision aid (updated version). Searches for updates to information in the first decision aid were conducted for all reports in the literature collected in the databases from 1 January 2018 to 31 May 2021. Searches for new information to be added were conducted for all time periods through 31 May 2021. Hand searches were also conducted as necessary. The keywords and search formulas used in the literature searches and the number of references extracted from the searches are listed (Table 1).

#### 2.2.2. Revised STEP 2 Contents (Table 2)

(1)Updated information

For the instrumental delivery item, we added the results of Shishido et al. (2018) [4] to those of Kurakazu et al. (2020) [9] and updated the number of instrumental deliveries with epidural anesthesia and natural birth to the information calculated using Review Manager 5 [10].

(2)Additional information

New information on maternal and infant outcomes was added regarding the effects of epidural anesthesia on mothers from birth to the postpartum period and the effects of epidural anesthesia on the unborn child.

The items related to mothers were cesarean section [9], amount of blood loss during birth [9], effects on lactation [11,12], frequency of maternity blues [9] and postpartum depression [13] (added information), and the effects on mother-child bonding [14].

For the items related to the child, information was added on the Apgar score [9], the rate of NICU admission [15], and the rate of autism spectrum disorder [16]. In addition, information was added to the table comparing the advantages and disadvantages of each method of epidural anesthesia and natural birth, as well as information on the impact on the child, the percentage of breastfeeding during hospitalization, and the percentage of maternity blues during hospitalization.

**Table 2 ijerph-20-06042-t002:** Included studies.

Outcome	First Version of the Decision Aid	Second Version of the Decision Aid
**Birth Outcomes**		
Emergency cesarean section	None	Kurakazu et al. (2020) [9]
Instrumental delivery	Shishido et al. (2018) [4]	Shishido et al. (2018) [4]
		Kurakazu et al. (2020) [9]
Total bleeding at birth of	None	Kurakazu et al. (2020) [9]
500 g or more		
**Postpartum outcomes**		
Breastfeeding	None	Orbach-Zinger et al. (2019) [11]
		Shishido et al. (2021) [12]
Maternity blues	None	Shishido et al. (2021) [12]
Postpartum depression	None	Uehara et al. (2021) [13]
Mother-infant bonding	None	Kano et al. (2021) [14]
**Infant outcomes**		
Apgar score	None	Kurakazu et al. (2020) [9]
NICU admission	None	Høtoft et al. (2021) [15]
Autism spectrum	None	Qui et al. (2020) [16]

#### 2.2.3. Revised STEP 3 Contents

STEP 3 is the part where the pregnant woman is “Setting clear priorities for decision-making”. Pregnant women choose how important they believe certain factors are, such as “Pain associated with delivery”, “Side-effects of epidural anesthesia”, “Effect on the delivery process”, “Effect on the baby’s health”, “Effect on breastfeeding”, and “Delivery costs”. Based on the items of information added in STEP 2, “Breastfeeding in infancy “, and “Increased chances of a baby being hospitalized in the NICU after birth” were added.

#### 2.2.4. STEP 1 and 4 Contents

STEP 3 is the part where the pregnant woman answers about “Learning how to make an informed decision”. STEP 4 is the part where the pregnant woman decides whether to choose epidural anesthesia or not. There were no changes or information added.

### 2.3. Study Participants

The eligibility criteria for participants were anesthesiologists and obstetricians with experience in performing epidural anesthesia, or midwives with at least three years of experience working in epidural anesthesia. The sample size was calculated based on previous study on face validity [17], and the number of participants was set to 5.

### 2.4. Data Collection

Participants were recruited using convenience sampling and snowball sampling methods.

The data collection period was from 15 October 2021 to 31 October 2022.

### 2.5. Outcomes

The outcomes of this study were the evaluation of face validity, including its appearance and clarity, and the evaluation of content validity.

### 2.6. Statistical Analysis

Basic statistics were performed for face validity and content validity of the decision aid with a frequency distribution using IBM SPSS Statistics (version 25.0; Static Base and Advanced Statistics, IBM Japan, Tokyo, Japan). Regarding the free-text data, after summarizing the free-text data, the data were described and divided into labels, and categories were extracted for each similarity and the data were interpreted.

### 2.7. Ethics

We first verbally explained our research using an explanatory booklet containing information about the provision of confidentiality and anonymity of their data. All participants gave their informed consent for inclusion before they participated in the study.

In conducting this study, consideration was given to the protection of human rights in compliance with the Declaration of Helsinki and the Ethical Guidelines for Life Sciences and Medical Research Involving Human Subjects. The Institutional Review Board of St. Luke’s International University, Tokyo, Japan approved the study protocol (21–A055, 15 October 2021).

## 3. Results

Five participants, two physicians (one obstetrician/gynecologist and one anesthesiologist), and three midwives were included in the study. The participating midwives had 13.7 mean years of clinical experience and 4.0 mean years of epidural anesthesia experience.

### 3.1. Evaluation of Face Validity

Booklet size (A4) was answered as “Appropriately sized” by all the participants (100%). Number of pages was answered as “Just the right length” by four participants (80%), and as “Slightly too long” by one participant (20%). Easier to read figures and tables was answered as “Strongly agree” by four participants (80%) and “Agree” by one participant (20%). Easier to understand the content was answered as “Strongly agree” by four participants (80%) and “Agree” by one participant (20%). Amount of information was answered as “Just the right amount” by all the participants (100%). Information balance was answered as “It’s a good balance of both” by two participants (40%), “Information is weighted in the direction of natural birth” by two participants (40%), and “Information is weighted in the direction of epidural anesthesia” by one participant (20%). For “Can you help pregnant women who choose a method of birth in the future? “and “Would you recommend a decision aid to pregnant women?”, all the participants (100%) answered “Agree” for both.

In each step of the evaluation, three participants (60%) answered “Extremely useful” and two participants (40%) answered “Moderately useful” to the question “How useful is STEP 1?”. All the participant (100%) answered “Extremely useful” to the question, “How useful is STEP 2?”. Four participants (80%) answered “Extremely useful” and one participant (20%) answered “Moderately useful” to the question, “How useful is STEP 3?” Four participants (80%) answered “Extremely useful” and one participant (20%) answered “Slightly useful” to the question, “How useful is STEP 4?”. Two participants (40%) answered “Extremely satisfied” and three participants (60%) answered “Satisfied” for the overall evaluation.

### 3.2. Free Description

Opinions expressed in the free response statements were described and divided into labels, and categories were extracted for each similarity in content through discussion between two researchers. The opinions of the participants obtained through free comments are indicated by “ ” and categories are indicated by ≪≫. There were 38 specific comments regarding the appropriateness of the content. These were classified into the following seven categories: “addition or revision of text”, “unification of expressions”, “need for explanation/information”, “lack of evidence”, “potential to mislead”, “questionable”, and “structure” (Table 3).

### 3.3. Overview of the Updated Aid (Appendix A)

The total number of pages in the second version of the decision aid was 44. One information item was updated (i.e., instrumental delivery), and eight items were added (i.e., cesarean section, amount of blood loss during birth, postpartum depression, bonding disorders, impact on lactation, Apgar score, NICU admission, and autism spectrum disorder).

### 3.4. Comparison with the International Patient Decision Aid Standards Instrument (Version 4.0)

Two researchers checked whether the second version of the decision aid met the international standard items of the International Patient Decision Aid Standards instrument (IPDASi; Version 4.0) [18]. The items that met the criteria were marked with a “0” and the items that did not meet the criteria were marked with an “x”. Additionally, the items that did not meet the second version of the decision aid were noted as “not applicable”. As a result of the evaluation, the second version of the decision aid is the Japanese version of IPDASi. The Japanese version of IPDASi (Version 4.0) met 6 out of 6 (Q1–Q6) qualification criteria, 6 out of 10 (C1–C6) accreditation criteria, and 15 out of 28 (QU1-QU13, QU15, QU20) quality criteria. Note that achievement of the qualification criteria requires a “yes” rating on all items, whereas achievement of the accreditation criteria requires a score of at least three points on each item.

## 4. Discussion

### 4.1. Evaluation of Face Validity

Responses regarding the figures and tables were positive in terms of the ease of viewing information. A study of 1062 adult Japanese participants reported that pictograms are more effective in conveying the gist of information than text [19]. In addition to textual explanations, the second version of the decision aid mainly uses pictograms to make information visually easy to grasp. Moreover, the second version of the decision aid was created using pie charts, bar graphs, tables, and other display methods selected according to the type of information, which we assume led to the easier readability of the decision aid.

There were mixed opinions on the balance of information. Of the participants, 40% stated that the information was biased toward natural birth. Based on a non-randomized controlled trial examining the effects of the first version of the decision aid on approximately 300 pregnant women, most of the pregnant women in the intervention group who were initially unsure of the birth mode in the pre-survey showed a willingness to choose epidural anesthesia in the post-test when using the first version of the decision aid [12]. This indicates that even if the information in the decision aid was slightly biased in the direction of natural birth, the decision aid did not significantly affect the pregnant women’s choice, and that the pregnant women were able to choose epidural anesthesia after learning about its risks. Additionally, the information that pregnant women wanted to know about epidural anesthesia included the effects of epidural anesthesia on the child, the effects of epidural anesthesia on birth, and the possibility of cesarean section [8]. This indicates that pregnant women sought information about the risks of epidural anesthesia. Therefore, the decision aid plays an important role in providing accurate information about risks related to the birth method, instead of only posting information on the balance of information. Based on the above, the balance of information may be somewhat biased in the direction of natural birth. However, the results of previous studies suggest that this is unlikely to affect the pregnant women’s decision-making. Thus, the face validity of this decision aid was confirmed.

### 4.2. Evaluation of Content Appropriateness

In “STEP2: Understanding the characteristics of the options”, there were several opinions that fell into the category of “I want more information/explanation”. The purpose of this report is to promote understanding of the characteristics of the options by providing comparative results from various perspectives, including the process of birth and the effects on the mother and newborn. As mentioned in the free comments, there was a possibility that pregnant women, who are the readers, would not have a good understanding of the options. Therefore, we considered adding a new Q&A section in the second version of the decision aid as a supplementary explanation of the results, as well as adding explanations focusing on topics that pregnant women are likely to have questions. Our review identified the following questions pertaining to three topics that pregnant women were likely to ask: (1) Why is instrumental delivery more common with epidural anesthesia? (2) Why is there more blood loss during birth with epidural anesthesia? (3) Why is the percentage of breastfeeding during postpartum hospitalization lower with epidural anesthesia? The reason for focusing on these three topics is that all of them are items for which there is a difference in the results when comparing natural birth and epidural anesthesia. We believe that pregnant women are likely to have questions about the difference in the results. In the Q&A section, we avoided using medical jargon as much as possible to make it easier for pregnant women to understand. Instead, we chose words that were easy to visualize while providing explanations. Kato et al. (2013) [20] investigated the effects of a pharmacist intervention using a Q&A pamphlet for migraine patients. The Qs in their pamphlet (questions regarding patient knowledge and behavior) were used to help patients reflect on their own knowledge and behavior. Such reflection was shown to increase the patients’ motivation to acquire knowledge and behavior they lack. This is believed to increase their motivation to acquire knowledge and behaviors that they lack. This tends to deepen the patients’ understanding of the subsequent A (explanation of answers), suggesting that the Q&A format of the pamphlet is useful in increasing awareness and understanding of the disease and drug therapy [20]. Therefore, we consider that by establishing a Q&A section and adding supplementary explanations to this decision aid, the anticipated questions of pregnant women are expected to be resolved and that their understanding of the information will be further deepened.

This study considers items that have changed significantly in content. Maternity blues is a transient mild depressive state that occurs after birth and often resolves spontaneously in less than two weeks. On the other hand, post-partum depression is a problematic mental condition of postpartum mothers. As the information on post-partum depression is based on a comparison of natural birth and epidural anesthesia, even if the information on maternity blues is deleted, there is sufficient information to compare the mental state of mothers after childbirth, and this would not pose a problem. Based on the above, the maternity blues item was deleted from the second version of the decision aid.

Regarding autistic spectrum disorder, many midwives are asked by pregnant women in midwifery outpatient clinics and antenatal check-up situations about the possible link between the method of birth and autistic spectrum disorder, as interest in developmental disabilities has been increasing in recent years. Therefore, the inclusion of new information in the decision aid will make the decision aid useful for midwives as an informational tool. On the other hand, the association between birth methods and autism spectrum disorder could have a very strong impact on pregnant women. In the second version of the decision aid, information on autism spectrum disorder was derived from the results of one study. However, as this item could be a factor that significantly influences pregnant women’s decision-making, we conducted another literature search to identify studies with a higher level of evidence. The literature search yielded three studies from PubMed: Wall-Wieler et al.’s (2021) cohort study of 123,175 subjects in Manitoba, Canada found no association between epidural anesthesia and autism spectrum disorder in children (adjusted HR 1.08, 95% CI 0.97–1.20) [21]. A Danish cohort study of 479,178 subjects also found no association between epidural anesthesia and autistic spectrum disorder (adjusted HR 1.05, 95% CI 0.98–1.11) [22]. In contrast, Hanley et al.’s (2021) [23] cohort study of 388,254 women in British Columbia, Canada found that epidural anesthesia was associated with a slight increase in autistic spectrum disorder (adjusted HR 1.09, 95% CI 1.00–1.15). However, we believe that further validation is needed as the authors stated that it is likely that all confounding factors have not been eliminated and that the data are not of high quality to show an association of epidural anesthesia with autistic spectrum disorder. Therefore, in the second version of the decision aid, Qiu et al.’s. (2020) results [16] showing an association between epidural anesthesia and autistic spectrum disorder were described. As autistic spectrum disorder is a great concern for pregnant women and may have a significant impact on their decision-making, and as evidence of research results on the association between epidural anesthesia and autistic spectrum disorder was also found to be insufficient, the autistic spectrum disorder section was revised to include a section on the association between the birth method and autistic spectrum disorder at this stage. Therefore, we decided to provide information in the form of amending the section on autistic spectrum disorder to state that no definitive conclusion has been reached at this stage regarding the association between the method of birth and autism. Therefore, we presume that the appropriateness of the content of this updated decision aid has been confirmed because of the revisions and additions that were made while adjusting the content.

Finally, IPDASi (Version 4.0) [18] was used to evaluate the eligibility criteria, accreditation criteria, and quality criteria. The eligibility criteria were classified as patient decision aids because all six items were met. As the criteria for certification were also met, it can be concluded that the updated decision aid has no risk of harmful bias. The quality criteria are met for 15 of the 28 items, but there is no cutoff. In future studies, we expect that field testing for pregnant women will increase the number of items that meet the quality criteria and improve the quality of the aid.

### 4.3. Clinical Implications

This updated decision aid is designed to be used by midwives as an informational tool when providing decision support to women. Therefore, we believe that midwives working in clinical settings must be provided opportunities for discussion and training on specific ways on how to use the updated decision aid as an informational and communication tool in health guidance and birth planning situations. It is anticipated that the decision-making support for women will be further enhanced by not only creating decision aids but also collaborating with clinical sites on how to utilize them.

It takes time for pregnant woman to read all of the decision aid developed in this study. Therefore, we suggest that it is necessary to improve the availability of pregnant women to obtain the information they want to find using Instagram from their smartphones.

### 4.4. Limitations

The research collaborators in this study were all physicians and midwives who are medical professionals. Therefore, as the study was conducted only from the perspective of medical professionals, it is necessary to consider the ease of understanding and comprehensibility for those who have experienced childbirth and pregnant women. In addition, the five doctors and midwives who participated in the study were not sufficient for the sample size. Thus, we consider that generalizations should be made with caution.

In the future, in accordance with the development process of a decision-making aid, it will be necessary to conduct field tests on pregnant women as a next step (i.e., conduct a Beta test: field test of feasibility to reflect the opinions of pregnant women who use the decision aid [24]), and to make further improvements to the decision aid.

## 5. Conclusions

The second version of the decision aid was updated with the latest information and additional information regarding maternal and infant outcomes, focusing on studies comparing natural birth versus epidural anesthesia. The decision aid was also created to meet the international standard for a decision-making aid [i.e., IPDASi (Version 4.0)].

The face validity of the second version of the decision aid was confirmed, as positive responses were received regarding its appearance and clarity. The appropriateness of the content was also confirmed from the results of the analysis of the free-text comments. Feld tests on pregnant women are warranted in the future.

## Figures and Tables

**Table 1 ijerph-20-06042-t001:** Search formulas based on keywords.

Content	Search Formula
① Epidural Anesthesia	((Analgesia, Obstetrical) OR (Anesthesia, Epidural)) AND
	(Labor, Obstetric)
	Cochrane Library Custom date range: 1 January 2018 to 31 May 2021
	PubMed Filters: from 2018–2021
	Cochrane Library: Cochrane Reviews 14, Cochrane Protocols 1
	PubMed 330
	((Analgesia, Obstetrical) OR (Anesthesia, Epidural)) AND
	(Labor, Obstetric)
	Filters: Meta-Analysis, Randomized Controlled Trial, Review,
	Systematic Review, Filters: from 2018–2021
	PubMed 107
② Decision Making and Epidural Anesthesia	(((Analgesia, Obstetrical) OR (Anesthesia, Epidural)) AND (Labor, Obstetric)) AND ((Decision Making) OR (Decision Aid)) Cochrane Library Custom date range: 1 January 2018–31 May 2021 PubMed Filters: from 2018–2021
	Cochrane Library: Cochrane Reviews 7 PubMed 15
③ Breast Feeding and Epidural Anesthesia	(((Analgesia, Obstetrical) OR (Anesthesia, Epidural)) AND (Labor, Obstetric)) AND (Breast Feeding)
	Cochrane Library Custom date range: to 31 May 2021
	PubMed Filters: to 31 May 2021
	Cochrane Library: Cochrane Reviews 4, Trials 17
	PubMed 42
④ Postpartum Depression and Epidural Anesthesia	(((Analgesia, Obstetrical) OR (Anesthesia, Epidural)) AND (Labor, Obstetric)) AND (Depression, Postpartum)
	Cochrane Library Custom date range: to 31 May 2021
	PubMed Filters: to 31 May 2021
	Cochrane Library: Cochrane Reviews 31, Cochrane Protocols 1, Trials 4 PubMed 19
⑤ Neonatal Outcomes and Epidural Anesthesia	(((Analgesia, Obstetrical) OR (Anesthesia, Epidural)) AND (Labor, Obstetric)) AND (Neonatal Outcomes)
	Cochrane Library Custom date range: to 31 May 2021
	PubMed Filters: to 31 May 2021
	Cochrane Library: Cochrane Reviews 74, Cochrane Protocols 3, Trials 97, Clinical Answers 1
	PubMed 479 (Filters: in the last 5 years: PubMed 102)
⑥ Child Development and Epidural Anesthesia	(((Analgesia, Obstetrical) OR (Anesthesia, Epidural)) AND (Labor, Obstetric)) AND (Child Development)
	Cochrane Library Custom date range: to 31 May 2021
	PubMed Filters: to 31 May 2021
	Cochrane Library: Cochrane Reviews 47, Cochrane Protocols 3
	PubMed 23

**Table 3 ijerph-20-06042-t003:** Corrected contents.

Comment	Revision
“addition or revision of text”	
There were six comments on the correct use of Japanese language.	Corrections were made to the correct the wording for six comments. ‘stick the injection’ (p. 57) was corrected to ‘inject’; ‘use epidural anesthesia’ (p. 60) to ‘choose epidural anesthesia’; and ‘rated as high in birth satisfaction’ (p. 68) to ‘defined as high in birth satisfaction’, respectively. In response to comments regarding the appropriateness of the wording considering the content and context, “advantages and disadvantages” (p. 59) was revised to “risks and benefits”; “1 in 4000 may exceed life-threatening complications” (p. 60) was revised to “very rarely”; in “(for urinary retention) very rarely does not improve for more than a week’ (p. 66), ‘very rarely’ was deleted”.
“unification of expressions”	
There were four comments on the expression.	Specifically, “local anesthetics” (p. 55) was changed to “anesthetics”; “cervix dilatation” (p. 56) to “full opening of the cervix”; and “tubes” (p. 57) to “thin tubes called catheters” to make the expressions clearer in the decision aid.
“need for explanation/information”	
Regarding the comparison table between natural and painless delivery (p. 59), one participant commented, “I thought it would be better to write about the effects on delivery (fever, prolonged delivery, suction delivery, etc.)	We have added a section entitled, ‘Impact on Childbirth’, for its relevance to the information provided in the decision aid.
**Comment**	**Revision**
With regard to the statement regarding how to spend the time during epidural anesthesia, “Since the patient cannot go to the toilet, urine is passed through a tube (catheter)”, one participant commented, “Generally, the patient cannot go to the toilet, but at our hospital, we explain that the patient goes to the toilet in a wheelchair”.	Assuming that each facility has a different situation, we added the information. Depending on the sensory and motor status of the lower extremities and the facility’s policy, the response may vary, such as walking on the toilet, using a wheelchair to move to the toilet, or inserting an indwelling bladder catheter.
Regarding the description of instrumental delivery (p. 63), the opinion that “The risks to the mother include a larger wound and a known increased risk of wound infection and third- and fourth-degree tears”.	We added an explanation of the risks of instrumental delivery to the mother.
Regarding STEP2 as a whole, “The lack of annotations for each result makes it difficult to interpret the results” and “Interpretation should be included”. Several participants commented that they found it difficult to understand the results without at least some explanation.	We added explanatory statements on pain (perineal pain) and fatigue related to childbirth, why epidural anesthesia is more likely to result in instrumental delivery and more blood loss than a natural birth, and the effect on breastfeeding, respectively. [Note: I do not see the need for using “respectively” here. Please recheck and remove if unnecessary]
Regarding “Use of drugs to relieve labor pains (including epidural anesthesia)” (p. 69), the opinion that “a footnote should be added to prevent confusion with epidural anesthesia”.	We have added specific descriptive information about the methods of relieving labor pains using drugs other than epidural anesthesia.
“lack of evidence”	
Concerning the data on lactation (p. 72), one participant commented, “There seems to be a bias, I don’t think we should insist on too much. If you are going to put it out there, it would be better to put out more domestic data on the next page”. Regarding the comparison chart between natural and painless delivery (p. 59), which included “Percentage of breastfeeding during hospitalization”, one participant commented, “I think it would be better not to include it in this section because a cause-and-effect relationship cannot be stated”.	Outside of Japan, and considering the use of aids in Japan, the content was modified by adopting domestic research papers and removed from the items in the comparison table.
“potential to mislead”	
Regarding the information on Maternity Blues (p. 74), a respondent stated, “I feel this is misleading. It seems to me that perhaps people who have epidural anesthesia = people who are nervous by nature, and there is no causal relationship between this and painless deliveries. If the causal relationship has been proven to some extent, I think it can be included individually, but if not, I think it should be removed from here”. On the other hand, one respondent commented, “I liked that the comparison between painless delivery and natural childbirth was expanded to include the effects on the mother after birth (maternity blues, postpartum depression, bonding disorders) and on the baby.”	After review, we considered that the level of evidence for maternity blues was low and potentially misleading, with only a small number of articles on maternity blues. We therefore removed this information from the second version of the decision aid.
**Comment**	**Revision**
Regarding the information on autism spectrum disorder (p. 80), one respondent commented, “I thought it would be useful as information that recent pregnant women would like to know about autism spectrum disorder, as I have the impression that more people are concerned about its relationship with epidural anesthesia or accelerated delivery at midwifery outpatient clinics”. On the other hand, “I thought it would be better to delete the description of ASD because it has quite a strong impact”, and “I wondered if the ‘autism spectrum disorder’ section was necessary and felt uncomfortable. It is unclear whether this is due to the wide range of disabilities and differences in delivery methods, or whether it is not related to cesarean sections, which are not included. This could be information that could stir up anxiety, so it is better to examine it carefully”. One respondent commented, “I think it would be better to examine the information”.	Once again, a literature search was conducted to gather information, and three studies were extracted from PubMed (search date: 8 November 2021). However, the results of one study showed slightly more autism spectrum disorder with epidural anesthesia than with natural birth, whereas the results of two studies showed no difference between natural birth and epidural anethesia, which means that the association between painless delivery and autism spectrum disorder is still unknown. Based on the above findings, we decided to provide information on autism spectrum disorder in the form of a correction to the content, stating that no conclusion has been reached on the relationship with the method of delivery at this stage, judging that the information is likely to be misleading owing to its strong impact on the reader and the lack of sufficient evidence.
“questionable”	
Regarding the reason for choosing the delivery method (p. 58), “I don’t understand how being fearful of pain is different from being afraid of pain. Is it because I am an older pregnant woman? I feel uncomfortable that the reason for choosing natural birth is not because it was recommended by a family member”, and “I wonder about ‘because I think natural birth is better’. If that is the case, I think the reason for choosing an epidural anesthesia should also be because I think epidural anesthesia is better”.	We revised the entire graph by reviewing the study data (Shishido, 2017) used in the reasons for the choice of delivery method and organizing the necessary data supplements and items that needed to be integrated.
Regarding the side effects of epidural anesthesia (p. 60), “This section tends to be misleading because it only mentions risks. Some commented, “I think it is essential to have information such as the cesarean section rate does not change, Apgar 5 min value does not change, etc.”	We did not make any modifications because we determined that organizing and providing information on the side effects of epidural anethesia (e.g., fever, hypotension, nausea, and vomiting) was necessary for choosing the delivery method.
Regarding birth satisfaction (p. 68), a participant said, “I think this could be quite misleading if it is not explained properly. Perhaps the pain-free people have too high expectations to begin with, or they have a different base character. I thought it would be better to make it a little milder”.	The design and color scheme were designed to emphasize that even among those who had epidural anethesia, those who were able to control their labor tended to have higher levels of satisfaction with their delivery.
“structure”	
Regarding the explanation of the general flow of vaginal delivery (p. 55) at the beginning of STEP 2, one respondent commented, “I thought it should come after ‘STEP 2: Knowing the characteristics of your options”.	We decided to keep the structure of the vaginal delivery process as it is described at the beginning of STEP 2 because it is a basic knowledge that is common to both natural birth and epidural anethesia.

## Data Availability

The data presented in this study are available on request from the corresponding author. The data are not publicly available due to privacy.

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
