# Peer review of "Updated Decision Aid Enabling Women to Choose between with or without Epidural Analgesia during Childbirth, and Confirmation of Validity"

_ijerph, 2023, doi:10.3390/ijerph20116042_

Round 1

Reviewer 1 Report

The manuscript needs to be more clearly written, and a study with 5 participants is irrelevant.

Author Response

We wish to express our deep appreciation to you for recognizing the importance of our study. Thank you for your favorable appraisal of the originality of the relevant data obtained. We also appreciate your valuable comments on the study introduction. We have responded to the main points that you mentioned in our point-by-point responses below. Thank you for the opportunity to clarify these points and make the necessary changes in the revised manuscript in accordance with your comments.

Reviewer 2 Report

This is a very thoughtful and well-written manuscript. 

  1. Expanding on affiliations to include city, state (if applicable), and country 
  2. Table #3: reorganize headers such that the content does not run into the next page - makes it easier for readers to follow
  3. Page numbers in Table #3 
  4. Ensure entire Table #3 is in the same font type and size
  5. The word ‘anesthesia’ is misspelled as “anethesia” and “aneshtesia” in some places in Table #3
  6. Line numbers are missing after the table 
  7. Under the header “Evaluation of content appropriateness,” the sentence: “However, as a result of adding various types of information, it became a list of information.” should be rewritten for clarity. 
  8. The sentence “However, the results of the three new studies extracted from the literature search either denied or indicated that further investigation is needed to show an association between epidural anesthesia and autistic spectrum disorder.” should be rewritten for clarity. “either denied or indicated” seems contradictory.

Author Response

(The authors gave the same response as above.)

Reviewer 3 Report

Dear Authors

Congratulation for your paper.

Interesting the approach of decision aid form analyze.

There are missing thinks in data's presentation.

Table 1- missing the period of search for points: 3,4,5,6

Row 129- basic statistics- need to be described the statistical method used in the paper

 Row 137- Ethics committee approval number.

In row 157-163 authors speaks about STEP1 to STEP 4 content.But are described only STEP 1 and STEP2.No information apart of this few rows about STEP3 and STEP 4.We cannot understand to which elements refer this 2 STEPS.

On pag 15 authors say that quality criteria and accreditation criteria are reached,but are not discussed which were this criteria's for quality  or accreditation.They have to be describe to understand the reader which were this ones.

At limitation paragraph -pag 18 the first part did not refer to limitation .Is a discussion not limitation of the study.This paragraph has to be more concise.

The length  of decision taking aid of 44 page I am afraid will be to long for pregnant woman to read it , understand it and take a decision.

I think a  shorter material will be more helpful for decision taking.

Author Response

(The authors gave the same response as above.)

Reviewer 4 Report

Notes on the article in the attached file.

Author Response

(The authors gave the same response as above.)

Round 2

Reviewer 1 Report

The manuscript can be accepted in the present format.